# PNF: Progressive normalizing flows

## Abstract

In this work, we introduce Progressive Normalizing Flows (PNF), a generative network that allows to model high-dimensional input data distributions by progressively training several flow-based modules. Competing generative models, such as GANs or autoencoders, do not aim at learning probability density of real data, while flow-based models realize this objective at a prohibitive cost of highly-dimensional internal representation. Here, we address these limitations and introduce a new strategy to train flow-based models. We progressively train consecutive models at increasing data resolutions, which allows to construct low-dimensional representations, as done in autoencoders, while directly approximating the log-likelihood function. Additional feature of our model is its intrinsic ability to upscale data resolution in the consecutive stages. We show that PNF offers a superior or comparable performance over the state of the art.

## 1. Introduction

Generative models trained to sample realistic data-points, such as images, are a mainstream field of machine learning, with multiple applications ranging from 3D shape optimization (Spurek et al., 2020) to high energy physics simulations (Deja et al., 2020).

Existing approaches include GANs (Goodfellow et al., 2014), autoencoders (Kingma & Welling, 2013) and invertible flows (Uria et al., 2014; Jain et al., 2020). GANs give high quality results, yet their main working principle relies on the competition between adversarially trained networks, while the sampling prior distribution is not explicitly modeled and hence does not provide an intuitive data sample representation. The second family of generative models, autoencoder-based methods, does not share

[1]Anonymous Institution, Anonymous City, Anonymous Region, Anonymous Country. Correspondence to: Anonymous Author <anon.email@domain.com>.

Preliminary work. Under review by INNF+ 2021. Do not distribute.

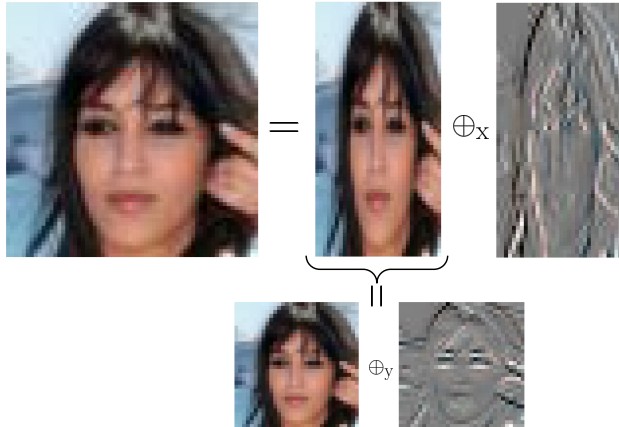

*Figure 1.* In PNF we start with a density model on a downscaled image. Next we we progressively add information encoded by a conditional flow model **(bottom images)** which allows us to upsample specific dimension of the image. In two steps of upscaling we upsample both dimensions, and finally obtain a high-resolution image **(upper left)**.

the limitation of GANs, as these models can simultaneously fit a data manifold and approximate its prior distribution (Kingma & Welling, 2013; Tolstikhin et al., 2017; Knop et al., 2020). More precisely, generative autoencoders map data into lower-dimensional latent space regularized to follow a certain distribution, *e.g.* Gaussian, and reconstruct it back. However, neither GANs nor autoencoder-based models are able to explicitly learn the probability density of real data in the input space.

The third family of generative models, flow-based methods, aims to address this limitation by constructing an invertible transformation from data space to a Gaussian distribution, called Normalizing Flow (Rezende & Mohamed, 2015). Contrary to the other methods, the model explicitly learns the data distribution, and therefore, the resulting loss function is defined as a negative log-likelihood. The main challenge in modeling flows is finding the invertible function that allows efficient computation of Jacobian determinant. For discrete flows, like NICE (Dinh et al., 2014), RealNVP (Dinh et al., 2016) and Glow (Kingma & Dhariwal, 2018) that issue is solved by application of the so-called *coupling layers*. The continuous flows, like FFJORD (Grathwohl et al., 2018) use Jacobian trace due to the transformation

specified by an ordinary differential equation.

Another important group of flow-based approaches are autoregressive methods, like IAF (Kingma et al., 2016) or MAF (Papamakarios et al., 2017) that uses chain rule and aims at modeling conditional distributions on single variables with an invertible transformations. Good approximation of negative log-likelihood, however, comes at a price of a high-dimensional internal representation which is costly both in terms of memory required to store the model, as well as the computational effort needed to train it.

In this work, we address these limitations of the existing generative models and introduce *Progressive Normalizing Flows* (PNF) model that takes the best of autoencoder and flow-based methods. Our approach combines the ability to construct low-dimensional representation, as done in autoencoder-based methods, with the direct approximation of the data distribution using negative log-likelihood function of flow-based models.

The main idea of PNF is to train several conditional flow generative models that learn the data distribution at recursively reduced resolutions, see Fig. 1. We use cascading multiple flow stages: the first flow model is trained using base resolution (*e.g.* $8 \times 8$ pixels), the second one is trained on a resolution increased twice along a given axis (*e.g.* $16 \times 8$), but with the conditioning mechanism taking the results of the base resolution model. This operation is recursively repeated until the model can output data samples in the desired resolution. To enable convergence of our model we represent data at subsequent stages by encoding the mean and divergence values, which allows to effectively train flow-based modules.

Our contributions can be summarized as follows:

- We introduce a divide-and-conquer progressive strategy to effectively train a high-dimensional flow-based generative models.
- We introduce a new PNF generative network that takes the best of two worlds of autoencoder and flow-based generative models: our approach constructs a low-dimensional data representation, while directly approximating the log-likelihood function.
- PNF has the latent given by images in lower resolution, which consequently allows the upsampling of images.

## 2. Description of PNF

**Basic idea of PNF**   We aim to define the generative model on high resolution images in a progressive fashion. In autoregressive models, we order the image pixels in a given way (Jain et al., 2020), and produce the next pixels by sampling from conditional distribution. In our paper, we apply a similar strategy, but we progressively increase the resolution

of the images. We sample higher resolution images from distribution conditioned on lower resolution versions.

To describe it, let us denote $\mathcal{I}_{ij}$ the set of images of resolution $2^i d \times 2^j d$ $(i, j = 0, 1, \ldots, k)$ and let us fix the image $M \in \mathcal{I}_k$ $(\mathcal{I}_k := \mathcal{I}_{kk})$. We denote by $M_{ij} \in \mathcal{I}_{ij}$ the image $M$ downscaled to the resolution $2^i d \times 2^j d$. Let us note, that one can obtain the image $M$ with the knowledge of the following information: the image $M_{00}$, the information necessary to upscale the resolution from $M_{ii}$ to get the image $M_{i,i+1}$, and from $M_{i,i+1}$ to $M_{i+1,i+1}$, for $i = 0, 1, \ldots, k - 1$. We will describe this idea more formally in the two following paragraphs. In the first, we give a detailed description in the case of one dimensional data (one-dimensional vector), and then we show how we can adapt it to the case of images (two-dimensional matrix).

**Theory: progressive approach in $\mathbb{R}^D$**   PNF is based on the fusion of the ideas standing behind autoregressive density models (Uria et al., 2014; Jain et al., 2020) and wavelet approach to data analysis and compression (in particular Haar wavelets).

Let us recall the basic idea of autoregressive models. We assume that our data lies in $\mathbb{R}^D$, and to model the density we apply the formula

$$p(x) = \prod_{i=1}^{D} p(x_i \,|\, x_1, \ldots, x_{i-1}), \quad x = (x_1, \ldots, x_D).$$

In our approach we use a version of autoregressive approach, but for a special decomposition of the space. To do this, we will need the following notation: given vectors $(x_1, .., x_n), (y_1, .., y_n) \in \mathbb{R}^n$ we define

$$x \oplus y = (x_1 + y_1, x_1 - y_1, \ldots, x_n + y_n, x_n - y_n) \in \mathbb{R}^{2n}.$$

For $x \in \mathbb{R}^{2n}$ we additionally define two operators

$$Sx := \left(\tfrac{x_1 + x_2}{2}, \ldots, \tfrac{x_{2n-1} + x_{2n}}{2}\right) \in \mathbb{R}^n,$$
$$\Delta x := \left(\tfrac{x_1 - x_2}{2}, \ldots, \tfrac{x_{2n-1} - x_{2n}}{2}\right) \in \mathbb{R}^n.$$

Observe that $S^k x$ is the point $x$ with decreased resolution, and that for $x \in \mathbb{R}^{2n}$ we have

$$x = Sx + \Delta x.$$

Applying the above formula $k$ times for $x \in \mathbb{R}^D$ $(D = 2^k d)$ we get

$$x = Sx \oplus \Delta x = (S^2 x \oplus \Delta Sx) \oplus \Delta x = \ldots$$
$$= [S^k x \oplus \Delta S^{k-1} x \oplus \ldots \oplus \Delta Sx] \oplus \Delta x.$$

Then we can apply the autoregressive density formula:

$$p(x) = C_1 \cdot p(\Delta x \mid Sx) \cdot p(Sx)$$
$$= C_2 \cdot p(\Delta x \mid Sx) \cdot p(\Delta Sx \mid S^2 x) \cdot p(S^2 x)$$
$$= \ldots = C_k \cdot p(S^k x) \cdot \prod_{i=1}^{d} p(\Delta S^{k-i} x \mid S^{k-i+1} x), \quad (1)$$

where $C_k$ is the inverse of th absolute value of the determinant of the map

$$\mathbb{R}^D = \mathbb{R}^k \times [\mathbb{R}^{2^0 d} \times \ldots \times \mathbb{R}^{2^k d}] \ni (v, v_0, v_1, \ldots, v_d) \to$$
$$\to [(v \oplus v_1) \oplus \ldots] \oplus v_d \in \mathbb{R}^D.$$

**PNF: model summary** Let us now summarize what is the final result of applying the conditional flow model to our data $X$ in the procedure described above. For $x \in \mathbb{R}^{2^l d}$ and $j < l$, by $x_{[j]} = S^{l-j} x \in \mathbb{R}^{2^j d}$ we denote the point $x$ "downscaled" to the resolution $2^j d$. Thus if by $\mathcal{X}$ we denote the true random vector from which our data $X$ was generated, by $\mathcal{X}_{[l]}$ we denote the rescaling of $\mathcal{X}$ to respective lower resolution. Then we obtain:

- invertible map $\phi : \mathbb{R}^d \to \mathbb{R}^d$, such that if $U \sim N(0, I)$, then $\phi(U) \sim \mathcal{X}_{[0]}$,

- for $j < l$ "upsampling" maps $\Phi_{jl} : \mathbb{R}^{2^j d} \times \mathbb{R}^{(2^l - 2^j) d} \to \mathbb{R}^{2^l d}$ such that if $x \sim \mathcal{X}_{[j]}$ and $U \sim N(0, I)$, then $\Phi_{jl}(x, U) \sim \mathcal{X}_{[l]}$.

Moreover, downscaling of $\Phi_{jl}(x, U)$ recreates $x$, i.e.

$$\Phi_{jl}(x, U)_{[j]} = x \text{ for } x \in \mathbb{R}^{2^j d}, \quad U \in \mathbb{R}^{(2^l - 2^j) d}. \quad (2)$$

Let us now observe that the above hierarchy gives us a latent model for $\mathcal{X}$. Namely by the latent space $Z$ we take $\mathbb{R}^d$, on which we sample by taking $\phi(U)$ for $U \sim N(0, I)$. Then the following mappings: encoder $\mathcal{E} : \mathbb{R}^D \to Z$ and decoder $\mathcal{D} : Z \to \mathbb{R}^D$, are given by

$$\mathcal{E}x = x_{[k]} \text{ and } \mathcal{D}z = \Phi_{0,k}(z, 0).$$

Observe that we upsample from $Z$ by taking the fixed zero noise. As always, the lower dimensional manifold spanned by $Z$ is given by $\mathcal{M} = \mathcal{D}Z$. Moreover, by (2) we obtain that $\mathcal{E}$ and $\mathcal{D}$ are right invertible, i.e.

$$\mathcal{D}\mathcal{E}z = z, \quad z \in Z.$$

In other words, we obtain that the following map

$$p_{\mathcal{M}} : \mathbb{R}^D \ni x \to \mathcal{D}\mathcal{E}x \in \mathcal{M}$$

is a true projection onto $\mathcal{M}$, that is

$$p_{\mathcal{M}}x = x, \quad x \in \mathcal{M}.$$

Observe that the above does not hold for the standard autoencoder models, as in the above we only obtain approximate identity.

**PNF for images** Now we describe the natural modification of the above approach for matrices (representation of images). As before, we consider $M \in \mathbb{R}^{D \times D}$, where $D = 2^k d$. Then we need the operators $S_x, S_y \, \Delta_x, \Delta_y$ and operations $\oplus_x, \oplus_y$, which are componentwise analogues of the operators $S, \Delta$ and operation $\oplus$. Notice that $S_x^i S_y^j M$ denote the image, where we reduce the resolution in $x$ by $2^i$, and in $y$ by $2^j$. Namely, we have

$$M = S_x M \oplus_x \Delta_x M = [S_y S_x M \oplus_y \Delta_y S_x M] \oplus_x \Delta_x M$$
$$= \ldots = [((S_y^k S_x^k M \oplus_y \Delta_y S_y^{k-1} S_x^k M) \oplus_y \ldots] \oplus_x \Delta_x M;$$

see Figure 1, where we depict the operations $\oplus_x$ and $\oplus_y$.

Then one can obtain the analogue of formula (1), which allows to compute the probability of the image $M$ in the original scale, by computing the probability of $M$ in reduced resolution multiplicated by the respective conditional probabilities which tell us "how much probabability" we have to add to increase the resolution.

Thus, exactly as is the case for vectors, PNF applied to images gives us the following features:

- density model on images,

- ability to upscale the resolution,

- lower dimensional manifold with the true projection.

## 3. Experiments

The main step in the training process of PNF is to train: $1°$ the baseline flow model $f_0$ modelling significantly reduced resolution images from the original dataset, $2°$ several conditional flow generative models which can be used in the process of upscaling the image resolution. We can use PNF approach appropriate number of times and finally compare, e.g., the resulting log-likelihood with the methods applied directly to high-resolution images from the original dataset. For the baseline generative flow model, we use *RealNVP*; see. (Dinh et al., 2016). We follow the same multi-scale architecture for all conditional models, as well.

For the toy-example, we choose *MNIST* dataset, upscaled to $32 \times 32$ pixels. Following (Dinh et al., 2016), we also consider *CIFAR-10* (Krizhevsky et al., 2009) and *CelebFaces Attributes* (Liu et al., 2015) (*CelebA*, downscaled to $64 \times 64$ pixels) datasets. To illustrate the point of our approach during the *training phase*, in the of case the original dataset with images of the size $D \times D$ ($D = 4d$), we proceed as follows. First we train the baseline RealNVP model $f_0$ for the downscaled images of the size $d \times d$. Next, we keep progressing by modelling four conditional flow models $f_1^x$, $f_1^y, f_2^x, f_2^y$ trained on the images rescaled to the resolution $2d \times d, 2d \times 2d, 4d \times 2d, 4d \times 4d = D \times D$, respectively. The training and conditioning observations are provided by application of $S_x, S_y$ and $\Delta_x, \Delta_y$ operators, described in

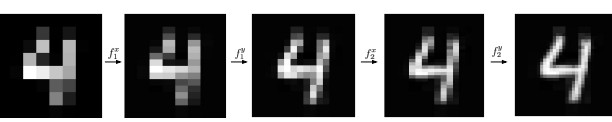

*Figure 2.* Upscaling pipeline with PNF models. From the left: image sampled by baseline *RealNVP* model $f_0$ trained on low-resolution ($8 \times 8$ pixels) *MNIST* dataset, upscaled images of the size $16 \times 8, 16 \times 16, 32 \times 16, 32 \times 32$ pixels with PNF models $f_1^x, f_1^y, f_2^x, f_2^y$, respectively.



*Figure 3.* Upscaling pipeline with PNF models. From the left: image sampled by baseline *RealNVP* model $f_0$ trained on low-resolution ($16 \times 16$ pixels) *CelebA* dataset, upscaled images of the size $32 \times 16, 32 \times 32, 64 \times 32, 64 \times 64$ pixels with PNF models $f_1^x, f_1^y, f_2^x, f_2^y$, respectively.

Section 2; see also Figure 1, where we depict the process of these operations in the case of test image from *CelebA* dataset.

For example, in the case of base images rescaled to $32 \times 32$ (*MNIST* and *CIFAR-10* case), we train the following models:

- baseline flow model $f_0$ for images $8 \times 8$,
- conditional flow models $f_1^x, f_1^y, f_2^x, f_2^y$ for images $\Delta_z(I)$ conditioned by $S_z(I)$ ($z \in \{x, y\}$), where $I$ is of the size $16 \times 8, 16 \times 16, 32 \times 16, 32 \times 32$, respectively.

Figure 2 shows a sample image generated by the above model $f_0$ and resulting upscaled images by using the generative models $f_1^x, f_1^y, f_2^x, f_2^y$; when upscaling the images we sample from generative flow models with a fixed zero noise.

In the case of *MNIST dataset* we consider both dense (linear) and convolutional architectures for RealNVP coupling layers. In the dense case, we use 6 affine coupling layers as invertible dense neural networks (18 layers). We follow (Dinh et al., 2016) and use the same architecture for convolutional coupling layers of *RealNVP* models for *CIFAR-10* (same for *MNIST*) and *CelebA* datasets. Figure 3 is analog of Figure 2, but with PNF trained on *CelebA* dataset.

For the purpose of comparison, for each considered dataset (images of $D \times D$ pixels), i.e. *MNIST* ($D = 32$), *CIFAR-10* ($D = 32$) and *CelebA* ($D = 64$), we train the reference baseline model $f$ and all the five PNF models $f_0, f_1^x, f_1^y, f_2^x, f_2^y$. We use regular train/test splits for each dataset. For the analysis, we compare the log-likelihood value (and resulting

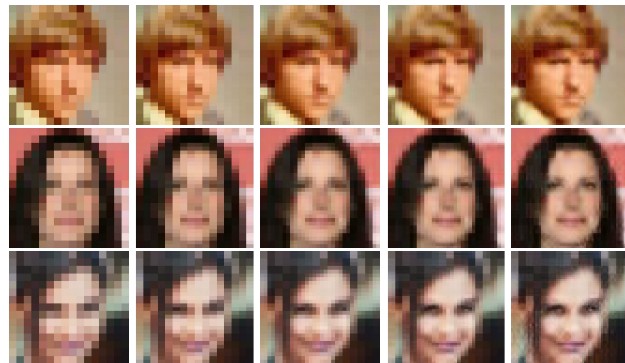

*Figure 4.* Upscaling the low-resolution *CelebA* images. Columns from the left: downscaled ($16 \times 16$ pixels) test *CelebA* image, upscaled images with PNF models $f_1^x, f_1^y, f_2^x$ and $f_2^y$, respectively.

*Table 1.* Log-likelihood and bits-per-dimension (in parenthesis) values for baseline RealNVP and our PNF models. For the *MNIST* dataset we considered both dense (top value) and convolutional (bottom) architecture of the coupling layers.

| DATA SET | REALNVP | PNF |
|---|---|---|
| MNIST | 4433.35 (1.75) | **4857.89** (**1.16**) |
| | 4473.32 (1.70) | 4264.79 (2.00) |
| CIFAR-10 | **9421.04** (**3.57**) | 8891.81 (3.82) |
| CELEBA | **46787.85** (**2.51**) | 44092.34 (2.82) |

*bits-per-dimension*) obtained by baseline model $f$ and all PNF models; we evaluate the models using testing split for each considered dataset. For the training process we chose *ADAM* algorithm (Da, 2014) with default hyperparameters and use an $L_2$ regularization on the weight scale parameters with coefficient $5 \cdot 10^{-5}$. We train all flow models by setting the prior $p_Z$ to be an isotropic unit norm Gaussian.

In the case of *CIFAR-10* and *CelebA* datasets the baseline models reproduces the results in (Dinh et al., 2016). Let us remark that, in the case of *CelebA* dataset, we trained the baseline model for more iterations and achieved better *bits-per-dimension* value (see Table 1) than the one given in reference paper (Dinh et al., 2016).

## 4. Conclusion

In this paper we introduced a new flow-based architecture, which can be seen as the fusion of classical flow models and autoregressive models. Since we split the space as done in the case of the wavelet transform, we obtain progressive densites of the data scaled to respective lower resolution. This allows us to view the lower resolution images as the latent, which is decoded to the original resolution by the upscaling given by the respective flow model with zero noise.

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
