# OpenReview forum: "PNF: Progressive normalizing flows"
_ICML.cc/2021/Workshop/INNF — Reject_

### Official Review · Reviewer_ELcG · 2021-06-07

**Rating:** Reject
**Confidence:** 5

**Summary:**

This paper presents a flow-based model framework with the aim to generate high-dimensional data.
The proposed approach takes the high-resolution data and progressively decreases its dimension by down-sampling.
This is done via operators similar to a Haar wavelet transform.
Then, using the chain rule for probability distributions, the approach gives a formula for modeling the high-resolution density based on low-resolution versions.
The approach is used for modeling image distributions such as MNIST, CIFAR-10, and 64$\times$64 CelebA.

**Justification For Rating:**



### Strengths and Weaknesses

+ This reviewer's main concern is that the general idea presented has already been investigated in prior work.
Closely related to this work, Wavelet-Flows [1] also try to use low-resolution versions of given images to enable high-resolution density estimation.
Similar to the proposed approach in this paper, Wavelet-Flows also use Haar transforms and the chain rule for probability distributions to enable high-resolution density modeling based on down-sampled versions of images.
Also, the idea of fusing Variational-AutoEncoders (VAE) with flow-based models has been investigated extensively, e.g., see [2-4].

+ Despite the aforementioned connections with prior work, this paper does not provide the reader with the information on how the proposed approach and its contributions might differ from existing methods.

+ To this reviewer, the paper is not easy to follow.
Please see the **additional feedbacks** for detailed comments/suggestions on making the paper clearer.

+ Finally, the experimental section lacks extensive investigation/discussion of the proposed approach.
Firstly, it only compares the proposed approach with Real-NVP [5] which is not a strong baseline.
Besides, the number of parameters in each approach is not mentioned.
Thus, it is not clear to this reviewer if the two models in Table 1 are comparable in terms of the number of parameters.
More importantly, from the results shown in Table 1, the proposed approach generally performs worse than the baseline in terms of log-likelihood.
Despite this, the results are not discussed thoroughly, and hence, it is not clear to this reviewer what the strengths of the proposed approach could be: does it have fewer parameters? does it require less training time? or maybe it generates higher-quality images in terms of FID?


### Additional Feedbacks and Suggestions

These are some of this reviewer's detailed suggestions to improve the current work clarity.

1.  In the paper, the term _autoencoder_ seems to be used instead of _variational autoencoder_ several times, e.g., see lines 016 and 042. Please consider using the exact term as it implies a different approach.

2. In the abstract, the paper reads: "competing generative models, such as GANs or [variational] autoencoders, do not aim at learning probability density of real data." This sentence does not exactly reflect the difference of GANs and VAEs with flow-based models. This is because both GANs and VAEs aim at learning the density of the real data. However, GANs do this _implicitly_, and VAEs only provide a lower bound to the _exact_ density. From the context, the paper seems to be trying to say that flow-based models differ from GANs and VAEs in that they provide an exact density function.

3. More precise references shall be used for "invertible flows" in line 042: for instance, see [6-8] below.

4. In the penultimate sentence of the first page, the paper reads: "For discrete flows, like NICE... ."
Consider using a different term instead of _discrete_ here.
What is meant by _discrete_ is in terms of not being represented by a continuous ODE model.
However, methods such as NICE still represent _continuous_ random variables.
The term _discrete_ in flow-based modeling usually refers to methods such as IDF [9] that model discrete random variables.
Please consider using other terms to avoid future confusion.

5. Coupling transformations are not the only way of efficient computation of the Jacobian determinant.
Consider altering the sentence "For discrete flows, ..." as currently the sentencing seems to claim that for all discrete-time normalizing flows this is the only approach.

6. Consider adding an in-depth discussion of the current literature (such as Wavelet-Flows) to clearly distinguish between the proposed approach and prior work.

7. While Figure 1 helps to understand the approach, adding a symbolic flow-chart is more beneficial.
To this end, consider adding a flow-chart reflecting the 2nd paragraph of Section 2 ("To describe it, let us...").

8. Consider using $\boldsymbol{x}$ or $\mathbf{x}$ instead of $x$ to distinguish between scalars and vectors.

9. The chain rule written in the paper can be problematic for $i=1$.
Consider replacing it with something more precise such as $p(\mathbf{x})=p\left(x_{1}\right) \prod_{i=2}^{D} p\left(x_{i} \mid x_1, \dots, x_{i-1}\right)$.

10. Currently, the use of $S\mathbf{x}$ and $\Delta\mathbf{x}$ is not well-motivated, and their introduction seems sudden.
Consider writing a few sentences justifying this move: how did you reach to the conclusion to use $S\mathbf{x}$ and $\Delta\mathbf{x}$?

11. The symbol $S^k\mathbf{x}$ is first used in line 098.
However, its definition is not entirely clear from the previous context.
Consider explaining what this symbol exactly denotes _just_ after using it, e.g., it denotes application of the operator $S$ on $\mathbf{x}$ for $k$ times.

12. The $+$ symbol in $\mathbf{x} = S\mathbf{x} + \Delta\mathbf{x}$ equation should be corrected to a $\bigoplus$ one.

13. Consider discussing the terms $C_k$ in Eq. (1) in more detail. For example, can it be computed exactly? How?

14. As stated above, consider a thorough discussion of the experimental results.
Given the results in Table 1, why one should opt to use the proposed approach?
What is exactly achieved?
Better quality images?
Less training time?
Less number of parameters?

15. The _Adam_ [10] optimizer is referenced mistakenly.
Consider correcting it.


#### References

[1] Yu, Jason J., Konstantinos G. Derpanis, and Marcus A. Brubaker. "Wavelet Flow: Fast Training of High Resolution Normalizing Flows." _NeurIPS_, 2020.

[2] Ma, Xuezhe, et al. "Decoupling Global and Local Representations from/for Image Generation." _ICLR_, 2021.

[3] Brehmer, Johann, and Kyle Cranmer. "Flows for simultaneous manifold learning and density estimation." _NeurIPS_, 2020.

[4] Lucas, Thomas, et al. "Adaptive density estimation for generative models." _NeurIPS_, 2019.

[5] Dinh, Laurent, Jascha Sohl-Dickstein, and Samy Bengio. "Density estimation using Real NVP." _ICLR_, 2017.

[6] Tabak, Esteban G., and Cristina V. Turner. "A family of nonparametric density estimation algorithms." Communications on Pure and Applied Mathematics 66.2 (2013): 145-164.

[7] Rezende, Danilo, and Shakir Mohamed. "Variational inference with normalizing flows." _ICML_, 2015.

[8] Dinh, Laurent, David Krueger, and Yoshua Bengio. "Nice: Non-linear independent components estimation." _ICLR Workshops_, 2015.

[9] Hoogeboom, Emiel, et al. "Integer discrete flows and lossless compression." _NeurIPS_, 2019.

[10] Kingma, Diederik P., and Jimmy Ba. "Adam: A method for stochastic optimization." _ICLR_, 2015.

---

### Official Review · Reviewer_KNZN · 2021-06-10

**Rating:** Borderline Reject
**Confidence:** 4

**Summary:**

The paper proposes a generative network that models the data by cascading low resolution data to higher resolution data. The basic idea of the method is described and experiments with RealNVP backbone are presented. Experimental results are reported for MNIST, CIFAR-10, and CelebA.

**Justification For Rating:**

The paper propose an engineering technique to improve normalizing flows. The claim that "PNF offers a superior or comparable performance over the state of the art" is inaccurate in my opinion. RealNVP is not state of the art for image generation anymore and still PNF only outperforms RealNVP when dense architectures are used.

Since this is an engineering paper, I think that not enough details about the architecture and training are provided, e.g., 1) how is low-resolution data obtained 2) how are cascading elements of PNF modeled in detail 3) are the results in Table 1 over a single run. Furthermore, I think the choices of the important upsampling operators (S and \Delta in paper) are not optimal.



The paper also misses crucial references to cascading models such as https://arxiv.org/abs/1812.01608 and https://arxiv.org/abs/1906.00446.

---

### Decision · Program_Chairs · 2021-06-14

**Decision:**

Reject

**Comment:**

The paper is on topic for this workshop. However, both reviewers are unsatisfied with the clarity of the method description, which also makes it very difficult to assess how this method differs from closely related (but not referenced) work such as Wavelet Flows. Furthermore, both reviewers think the baseline method is too weak and that therefore the claim for being comparable to the state of the art is too strong. Based on these reviews we have decided to reject this paper. We hope that a future submission will take into account the reviewer's feedback to improve the paper.